# Impact of Sodium Nitroprusside on the Photosynthetic Performance of Maize and Sorghum

**DOI:** 10.3390/plants13010118

**Published:** 2023-12-31

**Authors:** Georgi D. Rashkov, Martin A. Stefanov, Ekaterina K. Yotsova, Preslava B. Borisova, Anelia G. Dobrikova, Emilia L. Apostolova

**Affiliations:** Institute of Biophysics and Biomedical Engineering, Bulgarian Academy of Sciences, Acad. G. Bonchev Str., Bl. 21, 1113 Sofia, Bulgaria; megajorko@abv.bg (G.D.R.); martin_12.1989@abv.bg (M.A.S.); ekaterina_yotsova@abv.bg (E.K.Y.); preslavab12345@gmail.com (P.B.B.); aneli@bio21.bas.bg (A.G.D.)

**Keywords:** chlorophyll fluorescence, JIP parameters, nitric oxide, photosystem I, photosystem II, photosynthetic performance

## Abstract

Nitric oxide (NO) is an important molecule in regulating plant growth, development and photosynthetic performance. This study investigates the impact of varying concentrations (0–300 µM) of sodium nitroprusside (SNP, a donor of NO) on the functions of the photosynthetic apparatus in sorghum (*Sorghum bicolor* L. Albanus) and maize (*Zea mays* L. Kerala) under physiological conditions. Analysis of the chlorophyll fluorescence signals (using PAM and the JIP-test) revealed an increased amount of open PSII reaction centers (qP increased), but it did not affect the number of active reaction centers per PSII antenna chlorophyll (RC/ABS). In addition, the smaller SNP concentrations (up to 150 μM) alleviated the interaction of Q_A_ with plastoquine in maize, while at 300 μM it predominates the electron recombination on Q_A_Q_B_^−^, with the oxidized S_2_ (or S_3_) states of oxygen evolving in complex ways in both studied plant species. At the same time, SNP application stimulated the electron flux-reducing end electron acceptors at the PSI acceptor side per reaction center (REo/RC increased up to 26%) and the probability of their reduction (φRo increased up to 20%). An increase in MDA (by about 30%) and H_2_O_2_ contents was registered only at the highest SNP concentration (300 µM). At this concentration, SNP differentially affected the amount of P700^+^ in studied plant species, i.e., it increased (by 10%) in maize but decreased (by 16%) in sorghum. The effects of SNP on the functions of the photosynthetic apparatus were accompanied by an increase in carotenoid content in both studied plants. Additionally, data revealed that SNP-induced changes in the photosynthetic apparatus differed between maize and sorghum, suggesting species specificity for SNP’s impact on plants.

## 1. Introduction

Nitric oxide (NO) is a signaling molecule in plants involved in the regulation of many processes that plays an important role in plant growth and development under both physiological and stressful conditions [1,2,3]. NO is a small diatomic and highly reactive molecule with a high diffusivity exhibiting hydrophobic (lipophilic) properties [3,4]. Thus, NO easily migrates into a cell’s cytoplasm through the lipid phase of membranes and interacts with many receptors and signal molecules and scavenges ROS [2,4,5,6,7,8]. In this regard, NO has a role in plant signal transduction as a signaling factor in the cascade of events leading to changes in gene expression [2,4]. Over the course of the previous decades, attention has been paid to the mechanism of NO synthesis and its functions in plants. It has been established that in plants NO is generated through enzymatic (nitric oxide synthase and nitrate or nitrite reductases) and non-enzymatic pathways [3,9,10]. The endogenous production of NO takes place in different organelles like peroxisomes, mitochondria and chloroplasts [1,3,11]. Recent studies have shown that the endogenous content of NO in plants under physiological conditions is variety-specific [12].

It has also been demonstrated that NO plays a key role in controlling many processes in plants, including photosynthesis [1,11,13,14]. The influence of this signaling molecule on plants was investigated after their treatment with different NO donors [11,15]. The observed effects on the plants depend both on the type of donor and its concentration and the plant species. One of the most commonly used donors of NO is SNP. A previous study showed that low NO content influences the promotion of the biosynthesis of chlorophylls and proteins in the photosynthetic apparatus. This affects its efficiency, especially of the photosystem II (PSII) functions. The treatment with SNP of barley plants led to increased amounts of the light-harvesting complex of photosystem II (LHCII) that improves PSII photochemistry [16,17]. It has also been suggested that NO could participate in the stabilization of thylakoid membrane polypeptides. The electron transport pathway for photosynthetic processes can also be directly impacted by an exogenously applied NO donor [18,19]. The NO interacts with the acceptor side of the PSII complex (nonheme iron), with the tyrosine residue of D2 (YD) and the oxygen-evolving complex (OEC) being influences on the different oxidation states of the Mn clusters [1,19,20]. These interactions lead to changes in the PSII photochemistry. Spraying pepper and strawberry plants without very high concentrations of SNP has been shown to stimulate the photosynthetic activity of plants and improve their overall productivity [21,22]. While lower NO concentrations promote the biosynthesis of chlorophyll and proteins in the photosynthetic apparatus, higher NO contents could have negative effects on them [13,23,24]. It has been demonstrated that foliar spraying with 500 μM of SNP inhibits pigment accumulation and decreases photosynthetic capacity in tomato plants. It has been also shown that higher amounts of this signal molecule in plants accelerate leaf senescence [1]. The studies of different plant species have shown an increase in the closed PSII reaction centers and an influence on the photochemical energy conversion in PSII [14].

A considerable number of studies are related to the action of NO, in conditions of abiotic stress, of regulating the oxidative responses by enhancing the content of antioxidant enzymes and thus increasing abiotic stress tolerance in plants [1,12,25]. Since the effectiveness of NO donors is concentration-dependent [7,23,26] and can vary depending on plant species [12,27], more studies are needed to develop strategies for optimizing the application of NO donors in different forms of crop production under normal and stressful conditions.

Maize (*Zea mays* L.) and sorghum (*Sorghum bicolor* L.) are two of the most important cereals worldwide [28]. Although both can adapt well to a variety of global climates and can tolerate a wide range of environmental stresses, they have different types of sensitivity to environmental stress factors. In this study, the effects of different concentrations of SNP on the chlorophyll fluorescence at room temperature (PAM and the JIP-test), pigment composition, membrane stability and the levels of stress markers in sorghum (*Sorghum bicolor* L. Albanus) and maize (*Zea mays* L. Kerala) were investigated. We hypothesize that revealing the influence of NO donor SNP on the membrane stability and its impact on the various components of the electron transport chain of the photosynthetic apparatus will give more clarity on SNP’s action in thylakoid membranes as well as some of the reasons for different types of sensitivity in the plant species.

## 2. Results

### 2.1. Pigment Composition

Data revealed that in maize plants the total chlorophyll (Chl) content was unchanged after treatment with SNP concentrations of up to 150 µM and slightly decreased (by 5%) only after treatment with the highest studied concentration of SNP (300 µM) (Table 1). Furthermore, the carotenoid (Car) content increased (from 3% to 8%) after the treatment of maize plants with all the studied concentrations of SNP, except for the highest concentration (300 µM). The highest increase in the Car amount was observed after treatment with 50 µM of SNP (8%). The treatment with SNP of sorghum plants increased the content of Chl (from 4% to 15%) and Car (from 2% to 10%) more than the control leaves depending on the applied concentration of SNP. The highest increase in the Chl (15%) and Car (10%) was observed after treatment with 150 µM of SNP. This increase in Chl and Car contents was observed for all studied concentrations of SNP as the effect was stronger for the higher concentrations (150 and 300 µM SNP). At the same time, the experimental results revealed that no changes occurred in the Chl *a*/*b* and Car/Chl ratios in both studied plant species even after treatment with the highest studied concentration of SNP (Table 1).

### 2.2. Contents of Malondialdehyde and Hydrogen Peroxide

In both plant species, the highest applied concentration of SNP (300 µM) led to a slight increase in MDA (by about 30% in both species), while the increase in H_2_O_2_ contents was 45% in maize and 65% in sorghum (Figure 1).

### 2.3. Membrane Stability Index

The results showed that the values of the membrane stability index (MSI) were almost unchanged in comparison with the corresponding controls of both plant species, with an exception for maize plants treated with 300 µM of SNP (Figure 2) as the decrease at this concentration was about 2%. This parameter showed higher values (by about 3%) for all sorghum variants (0–300 µM SNP) compared with those for maize.

### 2.4. PAM Chlorophyll Fluorescence

The ratios of the quantum yields of photochemical and non-photochemical processes in PSII (Fv/Fo) and the photochemical energy conversion in PSII (Φ_PSII_) were only influenced in sorghum after treatment with SNP concentrations higher than 50 μM (Figure 3a,c). The parameter Fv/Fo decreased with 7%, while Φ_PSII_ increased (from 21% to 30%). At the same time, stimulation of the photochemical quenching (qP) compared with the controls was registered in both studied species after spraying with all SNP concentrations (Figure 3b). The increase observed for this parameter (qP) was larger in sorghum (from 24% to 31%) than in maize plants (from 7% to 11%) in comparison with the corresponding controls. At the same time, an increase in the excitation efficiency of open PSII centers (Φ_exc_) was observed only in sorghum after SNP treatment (Figure 3d).

The changes in the photosynthetic apparatus influenced the parameter R_Fd_ (correlating with the photosynthetic rate [29]) of the leaves of maize and sorghum differently. Data revealed a slight increase in maize plants after spraying with 150 and 300 µM of SNP (by 13%), while in sorghum this parameter was unchanged (Figure 4).

The kinetics of dark relaxation of chlorophyll fluorescence triggered through a single saturating light pulse of dark-adapted leaves for the characterization of Q_A_^−^ reoxidation [30,31] were also assessed. The fluorescence signal was fitted with two components (fast and slow) with rate constants of k_1_ and k_2_, respectively. These components characterized the different pathways of Q_A_^−^ reoxidation of that through plastoquinone and that through the electron recombination on the Q_A_Q_B_^−^ (via the Q_A_^−^Q_B_ ↔ Q_A_Q_B_^−^) charge equilibrium with oxidized S_2_ (or S_3_) states of the oxygen-evolving complex [32]. After the SNP treatment of up to 150 µM, the rate constant of the fast component (*k*_1_) increased in maize (from 18% to 21%), while in sorghum it was unchanged (Table 2). The slow rate constant (*k*_2_) showed high values for the variants treated with the highest SNP concentration (300 µM) of 25% in maize and 30% in sorghum. Moreover, SNP concentrations from 25 µM to 150 µM did not influence the constant *k*_2_ in maize, while in sorghum it decreased (from 13% to 40%) (Table 2).

### 2.5. Chlorophyll Fluorescence Induction

The chlorophyll fluorescence induction was used to provide more information about the influence of different levels of foliar application of SNP on the photosynthetic apparatus of maize and sorghum. The details for the JIP parameters used in this study are given in Materials and Methods. Data revealed some differences between studied control plant species (maize and sorghum) for parameters ABS/RC, ETo/RC, Vj, ψEo, Wk and RC/ABS (Figure 5). The results demonstrated that the values of parameters ABS/RC, ETo/RC, ψEo and Wk were higher in maize than in sorghum, while Vj and RC/ABS had a smaller value in maize compared with sorghum (Figure 5). The bigger differences between the two species were registered in parameters ETo/RC and ψEo that were higher in maize than in sorghum with 20% and 15%, respectively. The values of the parameters Vj and RC/ABS were higher in sorghum than in maize with 16% and 4%, respectively.

The foliar treatment with SNP concentrations above 50 µM led to an increase in the electron flux-reducing end electron acceptors at the PSI acceptor side (REo/RC—from 21% to 26%) and the quantum yield of their reduction (φRo—from 15% to 20%) in both studied species, while parameters φPo, Wk and RC/ABS had no influence (Figure 6). In addition, an increase in parameters ABS/RC, ETo/RC and ψEo in sorghum after SNP application was registered. Data also revealed small changes in parameter Vj (Figure 6).

Experimental results also revealed that untreated plants from both species were characterized by differences in the performance index PI_ABS_ (with higher value in maize), while the index PItotal (for energy conservation from exciton to the reduction of PSI end acceptors) was similar in maize and in sorghum (Figure 7). Furthermore, the parameter PItotal increased in treated maize plants with 150 and 300 μM of SNP (25–28%), while in sorghum it increased after the application of all the studied concentrations of SNP (from 13% to 24%) (Figure 7a). Additionally, the energy conservation from exciton to reduction in intersystem electron acceptors (PI_ABS_) was not influenced after SNP treatment in both studied species (Figure 7b).

The influence of SNP on the parameters involved in the determination of the performance indices are shown in Table 3. Data revealed that NO decreases the partial performance of primary photochemistry (parameter φPo/(1 − φPo)) in sorghum, while this parameter in maize was unchanged. Data also revealed that the performance of the thermal reactions of the intersystem electron carries [ψEo/(1 − ψEo)] was higher in maize than sorghum and also increased in sorghum after all the applied SNP concentrations, while in maize it diminished slightly at 25 and 50 µM of SNP. The parameter δREo/(1 − δREo) had higher values in untreated sorghum (1.073) plants than untreated maize (0.759) plants. In addition, this parameter increased after treatment with all the SNP concentrations in sorghum and 150 and 300 μM in maize (Table 3). The higher values of the parameter PItotal after SNP treatment in sorghum (all concentrations) and in maize (150 and 300 µM) were the result of the better efficiency of the electron transfer from Q_B_ to PSI electron acceptors [δREo/(1 − δREo)] (Figure 7a, Table 3).

### 2.6. Oxidation–Reduction Kinetics of P_700_

The impact of SNP on the photochemical activity of photosystem I (PSI) was evaluated by examining the redox characteristics of P_700_ following the far-red light-induced steady-state oxidation of P_700_^+^ (ΔA/A). The kinetics of P_700_^+^ dark reduction in both untreated and treated maize and sorghum plants were fitted through two exponential components with kinetic constants of *k_F_* and *k_S_*, respectively (Table 4). Results revealed that the fast constant (*k_F_*) was decreased only after treatment with 300 μM of SNP by 44% in maize and 27% in sorghum, while the slow constant (*k_S_*) was not influenced after all the SNP applications except for maize plants after 300 μM of SNP (by 29%) (Table 4). In addition, the changes in the ∆A/A ratio were not registered after spraying up to 150 μM of SNP in both studied plants (Table 4).

## 3. Discussion

Nitric oxide is a bioactive molecule that has an important role as a regulator of plant physiology and stress tolerance [1,33]. Despite the many studies on the influence of NO on plants, the molecular mechanisms of its action on the photosynthetic apparatus are not fully understood. In the present study, new experimental evidence for NO donor SNP’s effects on the photosynthetic apparatus of sorghum and maize are presented. The results showed that the SNP treatment caused different alterations in the photosynthetic pigments depending on the applied concentration and plant species, which is in agreement with previous investigations [13,17,23]. Our study revealed that SNP foliar spraying enhanced leaf Chl content (Table 1). This could be a result from the stimulation of Chl biosynthesis. A similar effect on Chl accumulation has been shown after treatment with 100 μM of SNP of barley seedlings [16] and of *Arabidopsis* [24]. Data also revealed an increase in Car content after SNP treatment in both studied species (Table 1). Carotenoids act as accessory light-harvesting pigments and also play an important role in photoprotection [34]. In addition, they are the integral component of the thylakoid membranes that also play an important role in the stabilization of the pigment-protein complexes of the photosynthetic apparatus [35,36,37,38]. It could be suggested that the stimulation of Car biosynthesis after SNP application is one of the reasons for the protective role of NO under environmental stress. Data demonstrated that the higher SNP concentration (300 μM) led to a decrease in the Chl amount in maize. This suggests a negative effect on Chl biosynthesis. A recent study also revealed an inhibition of Chl biosynthesis in tomato plants after applying a high SNP concentration (500 μM of SNP), inactivating the gene expression and translation of HY5 [13].

The experimental results also revealed that SNP treatment led to an increase in the stress markers (MDA and H_2_O_2_) at higher applied concentrations of SNP (150 and 300 µM), but their enhanced levels did not affect the membrane stability (MSI) except in maize plants treated with 300 μM of SNP (Figure 1 and Figure 2).

The lack of changes in membrane stability (MSI) independent of the increase in MDA and H_2_O_2_ can be explained through the SNP-stimulated accumulation of thylakoid membrane proteins [16] and the critical role of NO in regulating the stability of pigment-protein complexes in the thylakoid membranes [24]. In addition, NO had no influence on the ratio of Chl *a*/*b* that corresponded to the degree of thylakoid stacking [39,40]. Bearing this statement in mind, it could be suggested that NO does not influence the membrane stacking in both maize and sorghum. A similar influence of SNP on the Chl *a*/*b* ratio was observed after spraying the Iranian mandarin bakraii seedlings with 500 μM of SNP [41].

Chlorophyll fluorescence measurements were used to estimate the impact of SNP on the efficiency of the photosynthetic machinery. The data revealed an increase in the open PSII reaction centers (qP increase) after SNP treatment as the effect was more pronounced at concentrations of up to 150 μM for both species. This increase in the photochemical quenching (qP) was observed to a smaller extent in maize than in sorghum (Figure 3). The influence of SNP on the open PSII centers, alongside the increase in their efficiency, could also be a reason for the influence on the stress tolerance of the plants after exogenous treatment with SNP, a donor of NO [12]. At the same time, values for the rate constants (*k*_1_ and *k*_2_) of the decay of flash-induced variable fluorescence in both studied plant species were influenced in different ways (Table 2). These constants are related to Q_A_^−^ reoxidation through plastoquinone (*k*_1_) and through the recombination of electrons on Q_A_Q_B_^−^ with the oxidized S_2_ (or S_3_) state of the OEC (*k*_2_) [32]. The constant *k*_1_ increased in maize and *k*_2_ decreased in sorghum after treatment with a concentration of up to 150 μM of SNP. This suggests that SNP influenced the Q_A_^−^ reoxidation, i.e., it facilitated the electron flow from Q_A_ to plastoquinone in maize and influenced the recombination of electrons on Q_A_Q_B_^−^ in sorghum. The SNP-induced influence on the Fm decay kinetics suggests some changes in the acceptor side of PSII interacting with nonheme iron [20]. The impact of SNP on the parameter Vj also suggests the influence on the acceptor side of PSII (Figure 6). Current data also revealed that exogenous NO did not influence the numbers of active RC per PSII antenna chlorophyll (RC/ABS, Figure 6) regardless of the changes in chlorophyll content (Table 1) and the amount of LHCII [16]. Additionally, the impact on the PSII photochemistry influenced the electron flux-reducing end electron acceptors at the PSI acceptor side per RC (REo/RC) and the quantum yield of their reduction (φRo) (Figure 6). These changes were also connected with the performance index for energy conversion from photons absorbed by PSII to the reduction of PSI end acceptors (PI total) (Figure 7 and Table 3). It should be noted that larger values of the PI total after SNP application were the result from better efficiency of the electron transfer from Q_B_ to PSI electron acceptors [δREo/(1 − δREo)] (Table 3).

Analysis of the far-red light-induced oxidation of P_700_ and its dark reduction kinetics revealed an impact of SNP only at the highest concentration (300 μM) (Table 4). Previously, it has been suggested that both P_700_^+^ decay constants are associated with different populations of PSI complexes located in the stroma lamellae and the grana margin of the thylakoid membrane [42,43]. Therefore, these results suggest that treatment only with 300 μM of SNP influences the photo-oxidation of P_700_ in both populations of PSI in maize, while in sorghum this applies only to the PSI in the grana margin (i.e., the constant ks). In addition, a decrease in the ∆A/A ratio was observed only at the highest SNP concentration indicating an influence on the PSI complex in both species.

## 4. Materials and Methods

### 4.1. Plant Growth Conditions and SNP Treatment

In this investigation, maize (*Zea mays* L. Kerala) and sorghum (*Sorghum bicolor* L. Albanus) plants were used. The seeds were received from Euralis Ltd. (Lescar, France). The plants were grown in a half-strength Hoagland nutrient solution under controlled conditions that were as follows: 28 °C (daily)/23 °C (night) temperature, 150 µmol photons/m^2^ s light intensity, with a 12 h light/dark photoperiod and 60% air humidity. The two plant species were sprayed with different SNP concentrations (25 µM, 50 µM, 150 µM and 300 µM) after 14 days of the plant development. The effects of SNP were assessed after six days of leaf treatment.

### 4.2. Photosynthetic Pigments

The pigments were determined, as in [44], using ice-cold 80% acetone. The amounts of the total chlorophylls (Chl a + b) and carotenoids (Car) were measured spectrophotometrically using Specord 210 PLUS (Edition 2010, Analytik-Jena AG, Jena, Germany) and calculated using Lichtenthaler’s equations [45]. The pigment content in the leaves was calculated and presented as µg per g dry weight (DW).

### 4.3. Malondialdehyde and Hydrogen Peroxide Contents

The amounts of hydrogen peroxide (H_2_O_2_) and malondialdehyde (MDA) were estimated as described in Yotsova et al. [46]. The amounts of MDA and H_2_O_2_ were used to determine the levels of possible stress put on the plants [46,47]. The content of H_2_O_2_ was calculated through the absorption at 390 nm (Specord 210 Plus, Edition 2010; Analytik Jena AG, Jena, Germany) using the molar extinction coefficient of 0.28 µM/cm. The MDA amount was calculated through the absorbance at 532 nm and using the molar extinction coefficient 0.155 µM/cm. The amounts of MDA and H_2_O_2_ were expressed as nmoles per g DW.

### 4.4. Membrane Stability Index

Measurements of the cell membrane stability index (MSI) were performed to compare and assess the membrane stability of maize and sorghum leaves in response to foliar spraying with different concentrations of SNP. The MSI of leaf cells was evaluated by measuring the electrolyte leakage as described previously in Dobrikova et al. [48]. Mature leaves from different plants were cut into pieces and then incubated in plastic tubes with 40 ml of distilled water for 24 h at room temperature. The MSI values were calculated as MSI (%) = [1 − (EC1/EC2)] × 100 where EC1 and EC2 are the conductivity measurements before and after the boiling of the solutions with the leaf samples, respectively.

### 4.5. Chlorophyll Fluorescence Measurements

A fluorimeter (H. Walz, Effeltrich, Germany, model PAM 101–103) was used to measure the pulse-amplitude-modulated (PAM) chlorophyll fluorescence on mature leaves of maize and sorghum as in [47]. The leaves were dark-adapted for 20 min. The minimal chlorophyll fluorescence (Fo) value was obtained at 1.6 kHz instrumental frequency and with an intensity of a measuring pulse of 0.03 µmol photons/m^2^s. The maximal Fm (in dark condition) and Fm′ (in the presence of light) values were obtained using saturated light through 3000 µmol of photons/m^2^s (Schott lamp KL 1500 from Mainz, Germany). The actinic light intensity was 170 µmol of photons/m^2^s. The PAM chlorophyll fluorescence parameters used for the characterization of the effects of SNP on the photosynthetic apparatus functions were as follows [49,50]: Fv/Fo—the ratio of photochemical to nonphotochemical processes; qP—the coefficient of photochemical quenching; Φ_PSII_—the quantum yield of PSII photochemical energy conversion and Φexc—the excitation efficiency of open PSII centers [49,51]. The parameter R_Fd_ (the chlorophyll fluorescence decay ratio) was evaluated as in [47]. The rate constants of the decay kinetics of flash-induced variable fluorescence (*k*_1_ for the fast phase and *k*_2_ for the slow phase) in maize and sorghum leaves were determined as in [52]. The constants of *k*_1_ and *k*_2_ are associated with the pathways of Q_A_^−^ reoxidation [30,31]. The constant characterizes the interaction with plastoquinone, while *k_2_* characterizes the electron recombination on Q_A_Q_B_^−^ with OEC [32].

Chlorophyll fluorescence induction curves were measured using a Hansatech Handy PEA+ equipment (Norfolk, UK). The details for these measurements are given in [53]. The selected JIP parameters were used for the characterization of the impact of SNP on the photosynthetic apparatus as follows: the absorption flux per RC (ABS/RC), the electron transport flux from Q_A_ to Q_B_ per PSII (ETo/RC), the electron flux-reducing end acceptors at the acceptor side of PSI (REo/RC), the maximum quantum yield of primary photochemistry (φPo), quantum yield of reduction in end electron acceptors at the PSI acceptor side (φRo), the relative variable fluorescence at the J-step (Vj), movement of an electron into the electron transport chain beyond Q_A_^−^ (ψEo), the ratio of K-phase to J-phase (Wk), the numbers of active RC per PSII antenna chlorophyll (RC/ABS) and the performance indices PI_ABS_ (performance index (potential) for energy conservation from exciton to the reduction in intersystem electron acceptors) and PI total (performance index (potential) for energy conservation from exciton to the reduction in PSI end acceptors) [54,55,56]. Equations for performance indices are:PI_ABS_ = γRC2/(1 − γRC2) × φPo/(1 − φPo) × ψEo/(1 − ψEo);
PI total = PI_ABS_ × δREo/(1 − δREo)

### 4.6. Oxidation-Reduction Kinetics of P_700_

The redox characteristics of P_700_ were assessed on leaf discs using a PAM-101/103 fluorometer (manufactured by Walz in Effeltrich, Germany) with an attached ED-800T emitter-detector setup. The redox status of P_700_ was evaluated by detecting absorbance changes around 830 nm induced through far-red light (∆A/A). The dark reduction constants (*k_F_* and *ks*) of P700^+^, characterizing two different electron donor systems or two different pools of PSI located in different domains of thylakoid membranes, were also determined. These measurements were made as described in Stefanov et al. [47].

### 4.7. Statistics

The mean values (±SE) were calculated for two independent treatments with four measurements for each plant treatment. Statistically significant differences between the studied variants were identified using two-way ANOVA. Values of *p* < 0.05 were considered as statistically significant differences. For some of the results, statistical processing was carried out using the student’s *t*-test. Statistically significant differences between the compared values were marked (* *p* < 0.05, ** *p* < 0.01, *** *p* < 0.001).

## 5. Conclusions

In conclusion, the present study demonstrated the impact of foliar spray with SNP on the photosynthetic efficiency in maize and sorghum plants. The SNP treatment resulted in an increase in the open PSII reaction centers in sorghum (up to 31%) and in maize (up to 11%). The changes in the acceptor side of PSII alleviated the Q_A_^−^ reoxidation by plastoquinone in maize at SNP concentrations of up to 150 μM, while the treatment with the highest concentration (300 μM) predominated the electron recombination on Q_A_Q_B_^−^ (via the Q_A_^−^Q_B_ ↔ Q_A_Q_B_^−^ charge equilibrium), with oxidized S_2_ (or S_3_) states of the OEC observed in both plant species. In addition, SNP treatment increased the electron flux-reducing end electron acceptors at the PSI acceptor side per reaction center (about 26%) and the quantum yield of their reduction (about 20%). Despite changes in chlorophyll content, exogenous SNP did not influence the number of active reaction centers per PSII antenna chlorophyll. The changes in the photosynthetic apparatus correspond with an increase (up to 10%) in the Car content. The SNP-induced changes in the photosynthetic apparatus were observed to different degrees in maize and sorghum, suggesting species specificity for SNP’s impact on the plants. The observed various effects of SNP on the photosynthetic apparatus provide valuable insights into the role of NO in improving plant stress tolerance.

## Figures and Tables

**Figure 1 plants-13-00118-f001:**
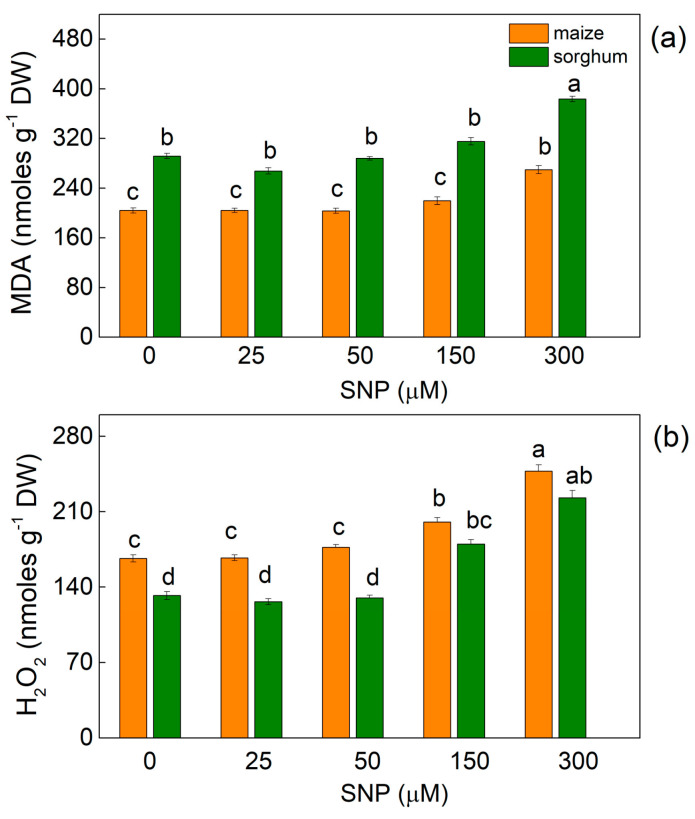
Effects of different SNP concentrations on (**a**) MDA and (**b**) H_2_O_2_ contents in the leaves of maize (*Zea mays* L. Kerala) and sorghum (*Sorghum bicolor* L. Albanus). Mean values ± SE (*n* = 8) are presented. Different letters indicate significant differences between the values (*p* < 0.05).

**Figure 2 plants-13-00118-f002:**
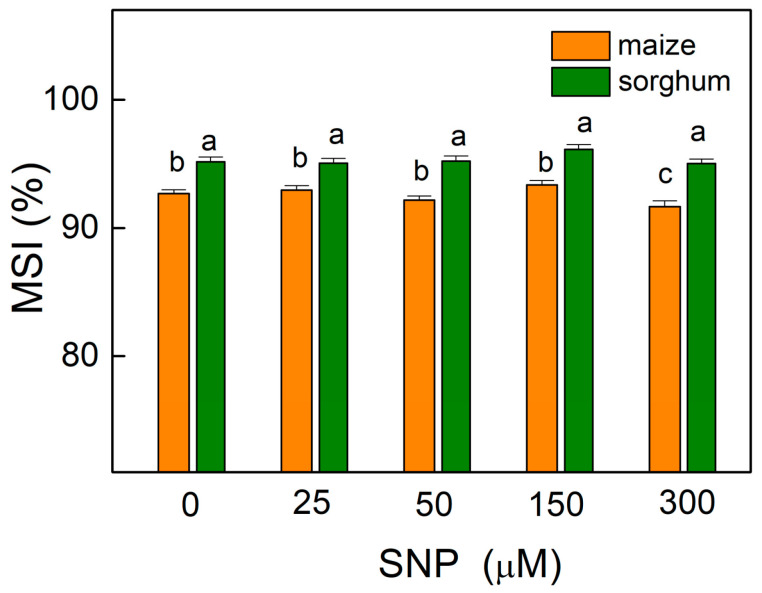
Effects of different SNP concentrations on the membrane stability index (MSI) in the leaves of maize (*Zea mays* L. Kerala) and sorghum (*Sorghum bicolor* L. Albanus) expressed in percentages from untreated plants. Mean values ± SE (*n* = 8) were presented and different letters indicate significant differences among treatments at *p* < 0.05.

**Figure 3 plants-13-00118-f003:**
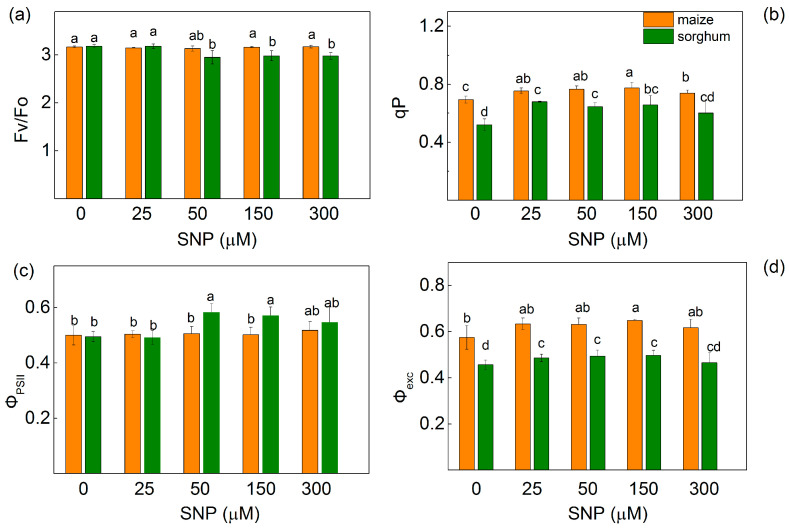
Effects of different SNP concentrations on PAM chlorophyll fluorescence parameters of maize (*Zea mays* L. Kerala) and sorghum (*Sorghum bicolor* L. Albanus) leaves. (**a**) The ratio of quantum yields of photochemical to non-photochemical processes (Fv/Fo); (**b**) photochemical quenching (qP); (**c**) effective quantum yield of photochemical energy conversion in PSII (Φ_PSII_); (**d**) excitation efficiency of open PSII centers (Φ_exc_). Mean values ± SE (*n* = 8) were presented and different letters indicate significant differences among treatments at *p* < 0.05.

**Figure 4 plants-13-00118-f004:**
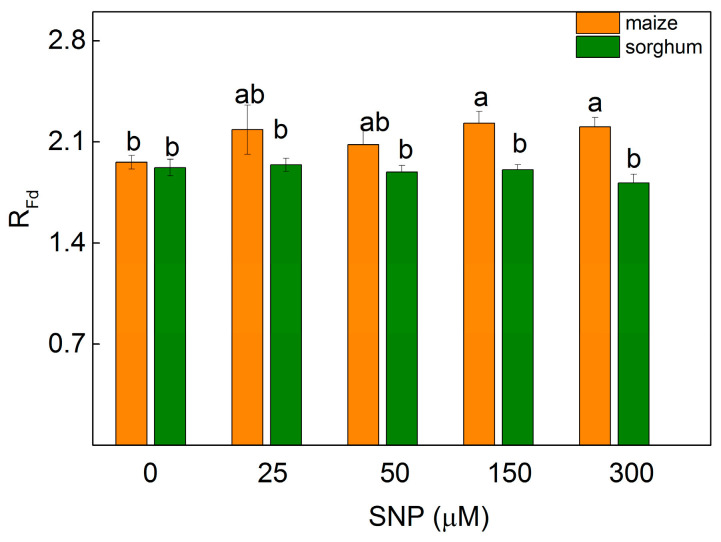
Effects of different SNP concentrations on the chlorophyll fluorescence decay ratio) (R_Fd_) on leaves from maize (*Zea mays* L. Kerala) and sorghum (*Sorghum bicolor* L. Albanus). Mean values ± SE (*n* = 8) are presented and different letters indicate significant differences among treatments at *p* < 0.05.

**Figure 5 plants-13-00118-f005:**
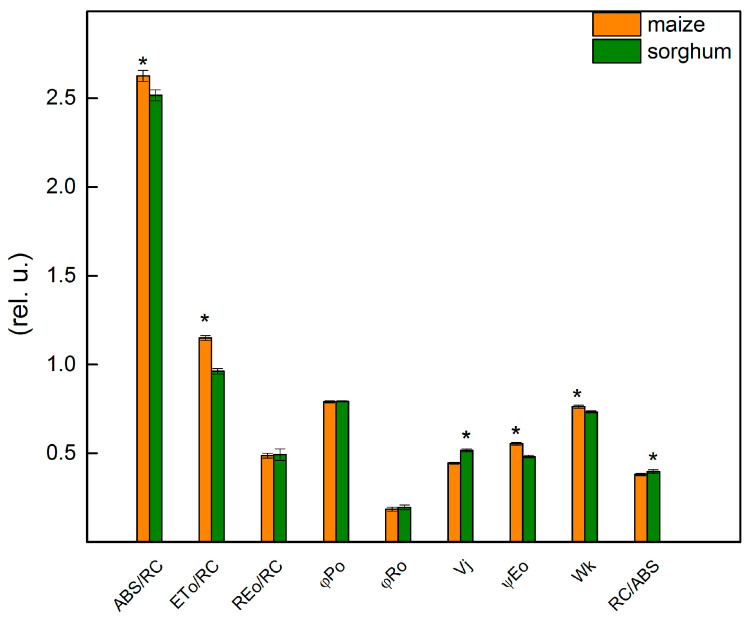
Selected JIP parameters of untreated plants from maize (*Zea mays* L. Kerala) and sorghum (*Sorghum bicolor* L. Albanus). ABS/RC (absorption flux per RC, apparent antenna size of an active RC); ETo/RC—electron transport flux (further Q_A_) per RC; REo/RC—electron flux-reducing end electron acceptors at the PSI acceptor side per RC; φPo—maximum quantum yield of primary photochemistry (at *t* = 0); φRo—quantum yield of reduction in end electron acceptors at the PSI acceptor side; Vj—relative variable fluorescence at the J-step; ψEo—movement of an electron into the electron transport chain beyond Q_A_; Wk—the ratio of the K-phase to the J-phase; RC/ABS—the number of active RC per PSII antenna chlorophyll. Mean values ± SE (*n* = 20) were presented and asterisks indicate the significant differences between the two plant species at *p* < 0.05.

**Figure 6 plants-13-00118-f006:**
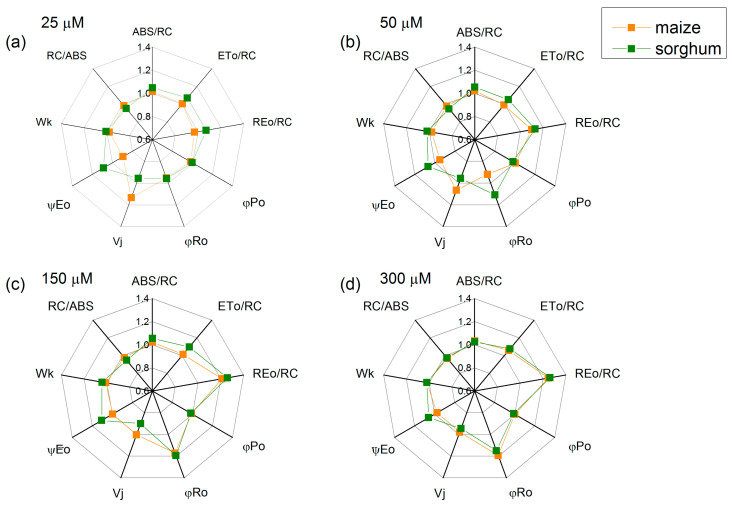
Effects of different SNP concentrations on selected JIP parameters in the leaves of maize (*Zea mays* L. Kerala) and sorghum (*Sorghum bicolor* L. Albanus) plants treated with (**a**) 25 μM of SNP, (**b**) 50 μM of SNP, (**c**) 150 μM of SNP and (**d**) 300 μM of SNP. ABS/RC (absorption flux per RC, apparent antenna size of an active RC); ETo/RC—electron transport flux (further Q_A_) per RC; REo/RC—electron flux-reducing end electron acceptors at the PSI acceptor side per RC; φPo—maximum quantum yield of primary photochemistry (at *t* = 0); φRo—quantum yield of reduction in end electron acceptors at the PSI acceptor side; Vj—relative variable fluorescence at the J-step; ψEo—movement of an electron into the electron transport chain beyond Q_A_; Wk—the ratio of the K-phase to the J-phase; RC/ABS—the numbers of active RC per PSII antenna chlorophyll. The parameters are normalized to the respective control. Mean values ± SE (*n* = 20) are presented.

**Figure 7 plants-13-00118-f007:**
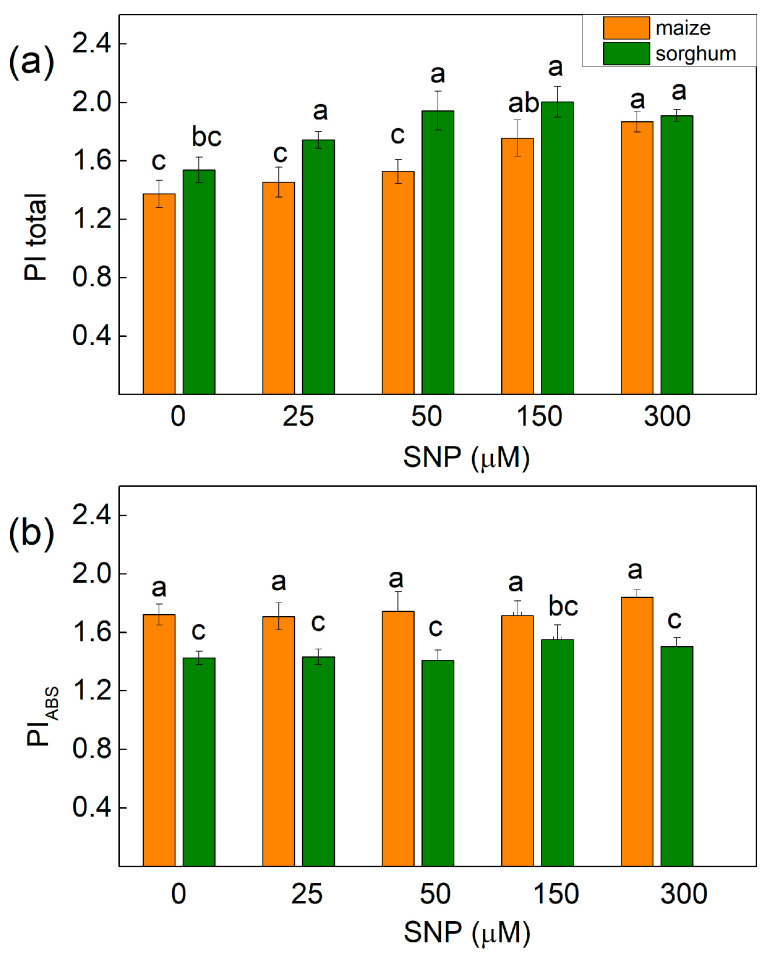
Effects of different SNP concentrations on (**a**) PItotal—the performance index for energy conservation from exciton to the reduction in PSI end acceptors and (**b**) PI_ABS_—the performance index for the energy conservation from exciton to reduction in intersystem electron acceptors in the leaves of maize (*Zea mays* L. Kerala) and sorghum (*Sorghum bicolor* L. Albanus). Mean values ± SE (*n* = 20) are presented and different letters indicate significant differences among treatments at *p* < 0.05.

**Table 1 plants-13-00118-t001:** Effects of different SNP concentrations on the pigment content in the leaves of maize (*Zea mays* L. Kerala) and sorghum (*Sorghum bicolor* L. Albanus). Mean values ± SE (*n* = 8) were presented as different letters that indicate significant differences between the values in the same column (*p* < 0.05).

Variants	Chl (µg/g DW)	Car (µg/g DW)	Chl *a*/*b*	Car/Chl
*Zea mays* L.				
0 µM SNP	28.03 ± 0.52 ^a^	7.91 ± 0.04 ^c^	4.25 ± 0.01 ^b^	0.21 ± 0.01 ^a^
25 µM SNP	28.66 ± 0.65 ^a^	8.30 ± 0.03 ^b^	4.26 ± 0.03 ^b^	0.22 ± 0.02 ^a^
50 µM SNP	28.00 ± 0.47 ^a^	8.56 ± 0.03 ^a^	4.25 ± 0.01 ^b^	0.24 ± 0.03 ^a^
150 µM SNP	27.93 ± 0.52 ^a^	8.16 ± 0.05 ^b^	4.44 ± 0.02 ^b^	0.22 ± 0.01 ^a^
300 µM SNP	26.74 ± 0.25 ^b^	7.86 ± 0.04 ^c^	4.36 ± 0.01 ^b^	0.22 ± 0.03 ^a^
*Sorghum bicolor* L.				
0 µM SNP	20.66 ± 0.12 ^e^	4.89 ± 0.03 ^g^	5.41 ± 0.02 ^a^	0.24 ± 0.02 ^a^
25 µM SNP	21.48 ± 0.06 ^d^	4.91 ± 0.02 ^g^	5.44 ± 0.02 ^a^	0.23 ± 0.03 ^a^
50 µM SNP	21.49 ± 0.30 ^d^	4.99 ± 0.03 ^f^	5.43 ± 0.03 ^a^	0.23 ± 0.02 ^a^
150 µM SNP	23.69 ± 0.57 ^c^	5.40 ± 0.02 ^d^	5.39 ± 0.01 ^a^	0.22 ± 0.01 ^a^
300 µM SNP	23.43 ± 0.67 ^c^	5.18 ± 0.04 ^e^	5.46 ± 0.01 ^a^	0.22 ± 0.01 ^a^

**Table 2 plants-13-00118-t002:** Effects of different SNP concentrations on the rate constants (*k*_1_ and *k*_2_) of the decay kinetics of the flash-induced variable fluorescence in leaves from maize (*Zea mays* L. Kerala) and sorghum (*Sorghum bicolor* L. Albanus). Mean values ± SE (*n* = 8) were presented and different letters indicate significant differences among treatments at *p* < 0.05.

Variants	*k*_1_ (s^−1^)	*k*_2_ (s^−1^)
*Zea mays* L.		
0 µM SNP	1.53 ± 0.14 ^b^	0.087 ± 0.003 ^b^
25 µM SNP	1.81 ± 0.15 ^a^	0.085 ± 0.015 ^b^
50 µM SNP	1.86 ± 0.07 ^a^	0.087 ± 0.006 ^b^
150 µM SNP	1.85 ± 0.09 ^a^	0.088 ± 0.003 ^b^
300 µM SNP	1.45 ± 0.08 ^b^	0.109 ± 0.012 ^a^
*Sorghum bicolor* L.		
0 µM SNP	1.87 ± 0.16 ^a^	0.092 ± 0.004 ^b^
25 µM SNP	1.84 ± 0.17 ^a^	0.080 ± 0.006 ^c^
50 µM SNP	1.94 ± 0.13 ^a^	0.069 ± 0.007 ^c^
150 µM SNP	1.85 ± 0.12 ^a^	0.055 ± 0.003 ^c^
300 µM SNP	1.76 ± 0.04 ^a^	0.120 ± 0.001 ^a^

**Table 3 plants-13-00118-t003:** Effects of SNP on the components of performance indices PI_ABS_ and PItotal in the leaves of maize (*Zea mays* L. Kerala) and sorghum (*Sorghum bicolor* L. Albanus). The parameters are in relative units. Mean values ± SE (*n* = 20) are presented and different letters indicate significant differences among treatments at *p* < 0.05.

Variants	γRC2/(1 − γRC2)	φPo/(1 − φPo)	ψEo/(1 − ψEo)	δREo/(1 − δREo)
*Zea mays* L.				
0 μM SNP	0.397 ± 0.004 ^a^	3.798 ± 0.057 ^a^	1.251 ± 0.032 ^a^	0.759 ± 0.044 ^c^
25 μM SNP	0.375 ± 0.007 ^b^	3.772 ± 0.089 ^a^	1.132 ± 0.051 ^b^	0.746 ± 0.059 ^c^
50 μM SNP	0.391 ± 0.007 ^a^	3.919 ± 0.089 ^a^	1.118 ± 0.051 ^b^	0.704 ± 0.059 ^c^
150 μM SNP	0.372 ± 0.009 ^b^	3.657 ± 0.095 ^a^	1.252 ± 0.056 ^a^	1.032 ± 0.066 ^b^
300 μM SNP	0.368 ± 0.007 ^b^	3.843 ± 0.097 ^a^	1.301 ± 0.058 ^a^	1.011 ± 0.078 ^b^
*Sorghum bicolor* L.				
0 μM SNP	0.397 ± 0.004 ^a^	3.844 ± 0.078 ^a^	0.933 ± 0.031^c^	1.073 ± 0.118 ^b^
25 μM SNP	0.379 ± 0.006 ^c^	3.564 ± 0.071^b^	1.027 ± 0.055^b^	1.307 ± 0.092 ^a^
50 μM SNP	0.388 ± 0.008 ^b^	3.549 ± 0.073 ^b^	1.019 ± 0.029^b^	1.234 ± 0.061 ^a^
150 μM SNP	0.376 ± 0.003 ^c^	3.562 ± 0.046 ^b^	1.154 ± 0.063^b^	1.281 ± 0.081 ^a^
300 μM SNP	0.388 ± 0.008 ^b^	3.615 ± 0.053 ^b^	1.046 ± 0.021^b^	1.269 ± 0.074 ^a^

**Table 4 plants-13-00118-t004:** Effects of different SNP concentrations on the relative amount of P_700_^+^ (∆A/A) and kinetic parameters of P_700_^+^ dark reduction (*k_F_* and *k_S_*) in the leaves of maize (*Zea mays* L. Kerala) and sorghum (*Sorghum bicolor* L. Albanus). Mean values ± SE (*n* = 8) were presented and different letters indicate significant differences among treatments at *p* < 0.05.

Variants	*k_F_* (s^−1^)	*k_S_* (s^−1^)	∆A/A
*Zea mays* L.			
0 μM SNP	0.321 ± 0.058 ^a^	0.031 ± 0.002 ^a^	10.70 ± 0.26 ^b^
25 μM SNP	0.363 ± 0.076 ^a^	0.030 ± 0.001 ^a^	10.55 ± 0.24 ^b^
50 μM SNP	0.384 ± 0.063 ^a^	0.035 ± 0.004 ^a^	10.70 ± 0.95 ^b^
150 μM SNP	0.341 ± 0.043 ^a^	0.030 ± 0.002 ^a^	10.91 ± 0.64 ^b^
300 μM SNP	0.180 ± 0.005 ^b^	0.022 ± 0.001 ^b^	11.79 ± 0.88 ^a^
*Sorghum bicolor* L.			
0 μM SNP	0.304 ± 0.047 ^a^	0.039 ± 0.003 ^a^	11.99 ± 0.55 ^a^
25 μM SNP	0.278 ± 0.070 ^a^	0.039 ± 0.004 ^a^	11.39 ± 0.34 ^a^
50 μM SNP	0.282 ± 0.034 ^a^	0.034 ± 0.004 ^a^	11.18 ± 0.43 ^a^
150 μM SNP	0.242 ± 0.047 ^ab^	0.030 ± 0.002 ^a^	10.85 ± 0.42 ^ab^
300 μM SNP	0.223 ± 0.025 ^b^	0.030 ± 0.003 ^a^	10.06 ± 0.23 ^b^

## Data Availability

Data are contained within the article.

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
