# Peer review of "Impact of Sodium Nitroprusside on the Photosynthetic Performance of Maize and Sorghum"

_plants, 2023, doi:10.3390/plants13010118_

Round 1
Reviewer 1 Report
Comments and Suggestions for Authors
Author Response
The authors would like to thank the reviewer for constructive and insightful comments about this work. We considered all comments and suggestions to be justified, and corrected the manuscript accordingly. Please, find the detailed list of all edits below. The newly edited text parts are indicated with red letters.
As per the title, the authors in the submitted article aimed to reveal the impact of nitric oxide on the photosynthetic performance in leaves of maize and sorghum. The study is interesting and certainly in the scope of the journal. The authors provide a detailed and qualitative description of the so far discovered impacts of nitric oxide (NO) on various elements of the photosynthetic machinery in the Introduction section. The article is easy to read, albeit it lacks quantitative assessment in many places even in the Introduction section (see comments later concerning the results), the biochemical and biophysical tools used here belong to the up-todate research methods. The paper needs significant improvement before it can be recommended for publication.
Below I summarize my concerns that need to be addressed.
One of the weaknesses of the current form of the manuscript is that the authors use sodium nitroprusside (SNP), a donor of NO, and interpret all the results as if the observed changes were exclusively due to the presence of NO. No quantitative assessment is presented on the efficiency of SNP to NO conversion under the conditions used. Is, for example, the production of NO dependent on the SNP concentration? If so, how? Unless the true NO content of the samples is identified, and the efficiency for the SNP to NO conversion is presented, the title should be changed by removing NO and adding SNP instead.
Answer: Thank you for the useful comment. We fully agree with this remark and in the revised manuscript we have removed NO and described the changes in the studied plants as a result of SNP treatment. We also made a change to the title of MS.
The authors use very uncertain (vague) terms that describe the relatively subtle observations in
the abstract and conclusion sections instead of quantifying them:
“…treatment with SNP influenced QA- reoxidation and increased photochemical quenching…”;
(line 15); “…stimulated the electron flux…”(line 17)
Answer: Some clarifications have been made in the Abstract and hope that the information is clearer in the revised manuscript (lines 14-19 and lines 20-24). More quantitative information about the influence of the SNP on the studied parameters is given in the Results.
“Significant impact of foliar spray…” (line 397); “changes in the acceptor side of PSII…” (line
400); “stimulation of the electron flux…” (line 401);
Despite of quantifying the changes, the authors again use a vague closing statement: The
observed various effects…..provide valuable information into the role of NO…(line 407).
These indeed would provide valuable information if the changes were specified and values
were attached to them (e.g. 15% increase/decrease, factor of two increase/decrease. etc.).
Answer: Your comments have been taken into account in the revised manuscript, the corrections and additions made are marked in red. Clarifications have been made in the Conclusions of the revised manuscript (lines 411-442).
In my opinion, readers must be provided with strong, assertive, and most of all, quantitative
statements on the accomplishments of the study. Figs. 2-4 display parameters that appear to
have very little variation on the concentration of SNP except at high concentrations (see the
next point) raising the concern whether they deserve to be displayed on dedicated figures.
On Fig. 1 the effect of SNP on malondialdehyde (MDA) and on H2O2 concentrations might raise some concerns. It appears that SNP has no effect on these parameters in the 0-50 ?? range and both parameters increase sharply at and above 150 ??. How did the authors exclude the possibility that the data displayed in Figs 2-4 at 150-300 ??concentration range contain the combined effect of SNP and MDA and/or H2O2? It appears that SNP has very subtle effect on the parameters displayed on these figures and thus, the authors emphasize these subtle changes.
Answer: Really our experimental results show an increase in the amount of MDA (by about 30%) and H2O2 (by 45% - 65%) at the highest concentration of 300 µM (Fig. 1). At the same time, there was no statistically significant changes in membrane stability (MSI, Fig. 2 ) except in maize treated with 300 µM SNP (about 1.5 % decrease). Based on previous studies showed that SNP stimulates the accumulation of thylakoid membrane proteins [16 from References] and the critical role of nitric oxide in regulating the stability of pigment-protein complexes in the thylakoid membranes [24 from References], we suggest that these increased amounts of the MDA do not cause significant inhibition of the functions of the photosynthetic apparatus (see also the explanation in the Discussion, lines 284-292). Moreover, our data revealed a small decrease (6 -7%) in the parameter Fv/Fo in sorghum treated with 50, 150 and 300mM SNP. On the other hand, data also revealed some changes in the acceptor side of PSII, which affect the QA reoxidation (Table 2), as after the treatment with 300mM SNP predominates the electron recombination on QAQB- (via the QA-QB ↔ QAQB- charge equilibrium), with oxidized S2 (or S3) states of the OEC in both plant species. On the other hand, the H2O2 could also serve as a signaling molecule activating defense mechanisms and is used to increase the plant tolerance under water deficit (Barzotto et al., Scientific Reports (2023) 13:13059 https://doi.org/10.1038/s41598-023-40388-y). The investigation of Gohari et al. [Physiol. Plantarum, 2019 doi:10.1111/ppl.13020] have suggested that the application of SNP and H2O2 might be an effective approach to improve the tolerance of aromatic plants to abiotic stress factors. The study of Hnilickova et al. [Plants, 2021, 10 845 https://doi.org/10.3390/plants10050845] have revealed that 4-fold increase of the MDA amount after salt stress leads to small changes in the parameter Fv/Fm. Therefore, we suggest that the increase in the amounts of MDA and H2O2 do not lead to damage to the thylakoid membranes and a decrease of the functions of the photosynthetic apparatus.
The physical processes that have been assigned to the fast and slow kinetic components of the flash-induced variable fluorescence should be identified.
Answer: The fast and slow components of the decay kinetics of the flash-induced variable fluorescence are assigned to the different pathways of QA reoxidation: by plastoquinone (fast component and constant k1) and recombination of electron on QAQB- (via the QA-QB ↔ QAQB-) charge equilibrium, with oxidized S2 (or S3) states of the oxygen-evolving complex (slow componets k2) [Ref. 30,31,32 in the MS] (lines 157-163 in the MS).
Sincerely yours,
Dr. Emilia Apostolova

Reviewer 2 Report
Comments and Suggestions for Authors
Dear Authors
Greetings!!!!
I have read your manuscript entitled “Impact of Nitric Oxide on the Photosynthetic Performance of Maize and Sorghum” with full interest and I found that the manuscript is very well written, and all the sections (Introduction, Material Method, Discussion) are described correctly and nicely by the authors, the overall quality of the manuscript is good. So, I do not have many major comments, and the manuscript can be accept in its present form
Author Response
The authors thank for the positive comments on the manuscript. Some corrections have been made to the manuscript in relation to the suggestion of the other two reviewers
Sincerely yours,
Dr. Emilia Apostolova
Reviewer 3 Report
Comments and Suggestions for Authors
Dear Authors,
the aim of you study is very interesting but, the quality of the results description is very low and often is not correct and this influenced negatively the quality of the other sections "discussion and conclusion. I wrote some comments on the manuscript that I hope could help you to re-write completely the manuscript for a new submission. At actual status, this manuscript must not be considered for publication and you need to re-write it completely, giving less attention to the introduction that seems to be well written

Author Response
The authors would like to thank the reviewer for constructive and insightful comments in relation to this work. We considered all comments and suggestions to be justified, and corrected the manuscript accordingly. Please, find the details of all edits in the fail. The newly edited text parts are indicated with red letters.
Sincerely yours,
Dr. Emilia Apostolova

Round 2
Reviewer 1 Report
Comments and Suggestions for Authors
The manuscript was improved somewhat. I do have still concerns. However, I do not want to delay the evaluation process any longer. Let the readers decide the value of the paper.
Suggestions: Please include the assignment of the kinetic components in the paper (materials and methods section).
Author Response
Dear Reviewer,
Thanks for the suggestions, which were taken into consideration in the revised manuscript.
Sincerely yours,
Emilia Apostolova